# Subdued, a TMEM16 family Ca²⁺-activated Cl⁻ channel in *Drosophila melanogaster* with an unexpected role in host defense

Xiu Ming Wong[1], Susan Younger[2,3], Christian J Peters[2], Yuh Nung Jan[2,3], Lily Y Jan[2,3]*

[1]Graduate Program in Chemistry and Chemical Biology, University of California, San Francisco, San Francisco, United States; [2]Department of Physiology, University of California, San Francisco, San Francisco, United States; [3]Howard Hughes Medical Institute, University of California, San Francisco, San Francisco, United States

**Abstract** TMEM16A and TMEM16B are calcium-activated chloride channels (CaCCs) with important functions in mammalian physiology. Whether distant relatives of the vertebrate TMEM16 families also form CaCCs is an intriguing open question. Here we report that a TMEM16 family member from *Drosophila melanogaster*, Subdued (CG16718), is a CaCC. Amino acid substitutions of Subdued alter the ion selectivity and kinetic properties of the CaCC channels heterologously expressed in HEK 293T cells. This *Drosophila* channel displays characteristics of classic CaCCs, thereby providing evidence for evolutionarily conserved biophysical properties in the TMEM16 family. Additionally, we show that knockout flies lacking *subdued* gene activity more readily succumb to death caused by ingesting the pathogenic bacteria *Serratia marcescens*, suggesting that *subdued* has novel functions in *Drosophila* host defense.

## Introduction

TMEM16A (*Caputo et al., 2008*; *Schroeder et al., 2008*; *Yang et al., 2008*) and, a different family member, TMEM16B (*Pifferi et al., 2009*) encode the classic calcium-activated chloride channels (CaCCs) in various mammalian tissues. Previously observed in a variety of organisms from green algae (*Fromm and Lautner, 2007*; *Shiina and Tazawa, 1987*) to *Xenopus* (*Miledi and Parker, 1984*; *Kline, 1988*), these channels are activated by an increase in cytosolic calcium with outward rectification at low calcium levels, and they preferentially permeate larger anions (*Large and Wang, 1996*; *Qu and Hartzell, 2000*).

In mammals, TMEM16A regulates fluid secretion in submandibular glands, (*Yang et al., 2008*; *Romanenko et al., 2010*) as well as on the epithelia of airway surfaces (*Rock et al., 2009*). This channel also modulates arterial (*Manoury et al., 2010*; *Bulley et al., 2012*), tracheal (*Huang et al., 2012a*), and gastrointestinal smooth muscle tone (*Hwang et al., 2009*), and has been observed to play a role in noxious heat sensing in the peripheral nervous system (*Cho et al., 2012*). TMEM16B is expressed in photoreceptor terminals (*Stohr et al., 2009*), where CaCCs are hypothesized to stabilize presynaptic membrane potential (*Lalonde et al., 2008*). This channel also gives rise to the majority of recorded CaCC current in hippocampal pyramidal neurons (*Huang et al., 2012b*) and olfactory sensory neurons (*Billig et al., 2011*).

Besides TMEM16A and B, only one other mammalian family member, TMEM16F, has been biophysically characterized in vitro and in vivo (*Yang et al., 2012*), which begs the question of whether or not the rest of the mammalian TMEM16 family encodes CaCCs (like TMEM16A and B) or small-conductance calcium-activated non-selective cation (SCAN) channels (like TMEM16F).

Outside of mammals, even less is known about the TMEM16 family. Ubiquitous in eukaryotes, TMEM16 family members regulate a bewildering variety of physiological functions. Ist2p, the single ortholog of the TMEM16 family in *Saccharomyces cerevisiae*, has been shown to function in endoplasmic

*For correspondence: Lily.Jan@ucsf.edu

**Competing interests:** The authors declare that no competing interests exist.

**Reviewing editor**: Utpal Banerjee, University of California, Los Angeles, United States

**eLife digest** Ions are at the root of most processes that occur in the body, so they must be able to move in and out of cells. However, because they have an electric charge, ions are not usually able to pass through the fatty membrane that encloses all cells. Instead, they must be imported or exported by a variety of dedicated proteins in the cell membrane. These include ion channels – proteins that, under certain conditions, open to form pores – and ion transporters.

Calcium-activated chloride channels are ion channels that permeate chloride ions when the concentration of calcium ions inside the cell increases. Two important calcium-activated chloride channels in mammals belong to the TMEM16 family of proteins, which is conserved in many organisms. However, to date all the examples of TMEM16 proteins forming calcium-activated chloride channels have been found in vertebrates. Moreover, it is not known how many members of the TMEM16 family can act as ion channels. Wong et al. have now isolated a protein belonging to the TMEM16 family from fruit flies and, in a series of experiments on human cells, showed that it acts as a calcium-activated chloride channel.

Previous work has shown that fruit flies lacking this protein, which is called Subdued, are more susceptible than wild-type flies to a pathogenic bacterium called *Serratia marrescens*, which implies that the Subdued ion channel might be involved in the immune system. Indeed, Wong et al. found that the mutant flies died more often than wild-type flies after eating these bacteria; the mutant files also had higher levels of the bacteria in their digestive tracts. These results will be of interest to researchers trying to understand how TMEM16 ion channels evolved to be involved in processes as diverse as vision, the secretion of bodily fluids and the immune system.

reticulum–plasma membrane tethering (*Wolf et al., 2012*). A member of the TMEM16 family in *Drosophila*, Axs, is found on the meiotic spindle and regulates meiotic chromosomal segregation (*Kramer and Hawley, 2003*). *Xenopus* TMEM16A functions to block polyspermy in fertilized oocytes and is to date the only non-mammalian TMEM16 member described as a CaCC (*Schroeder et al., 2008*). We thus have a limited understanding of both the biophysical and functional aspects of the TMEM16 family and whether these properties are evolutionarily conserved.

In an attempt to uncover TMEM16 family members with CaCC or SCAN channel activity, we cloned and heterologously expressed TMEM16 members from various genetically tractable organisms for electrophysiological inspection. A *Drosophila melanogaster* TMEM16 ortholog, CG16718, was found to be a CaCC upon heterologous expression in HEK 293T cells. In addition, we observe that this channel plays a role in host defense in *Drosophila*, a function that has not been previously reported in the TMEM16 family. Given the physiological role of this newly identified CaCC, we chose to give CG16718 the name *subdued*.

## Results

### Subdued is an ortholog of mammalian TMEM16A and B

Multiple sequence alignment with the mammalian TMEM16 family shows that Subdued is most similar to TMEM16A and B (*Figure 1A,B*), sharing 32.8% and 31.7% identity with these channels, respectively.

Heterologous expression of Subdued would have been ideally done in the commonly used *Drosophila*-derived S2 cell line. However, this cell line was reported to robustly express bestrophins that give rise to endogenous calcium-activated chloride currents (*Chien et al., 2006*), potentially confounding the analysis. Alternatively, Subdued was expressed in human embryonic kidney (HEK) 293T cells, a common heterologous expression system for electrophysiological studies of CaCCs (*Caputo et al., 2008*; *Schroeder et al., 2008*; *Yang et al., 2008*).

### Subdued is a calcium and voltage-dependent channel

48 hr post-transfection, HEK 293T cells expressing Subdued were recorded from using whole-cell patch clamp. Upon activation with 20–200 µM free intracellular calcium and symmetric NaCl solutions, the cells showed large time-dependent currents. In zero-calcium pipette solutions, no currents were observed in the transfected cells (*Figure 2A*). For this study, all recordings were done with 200 µM

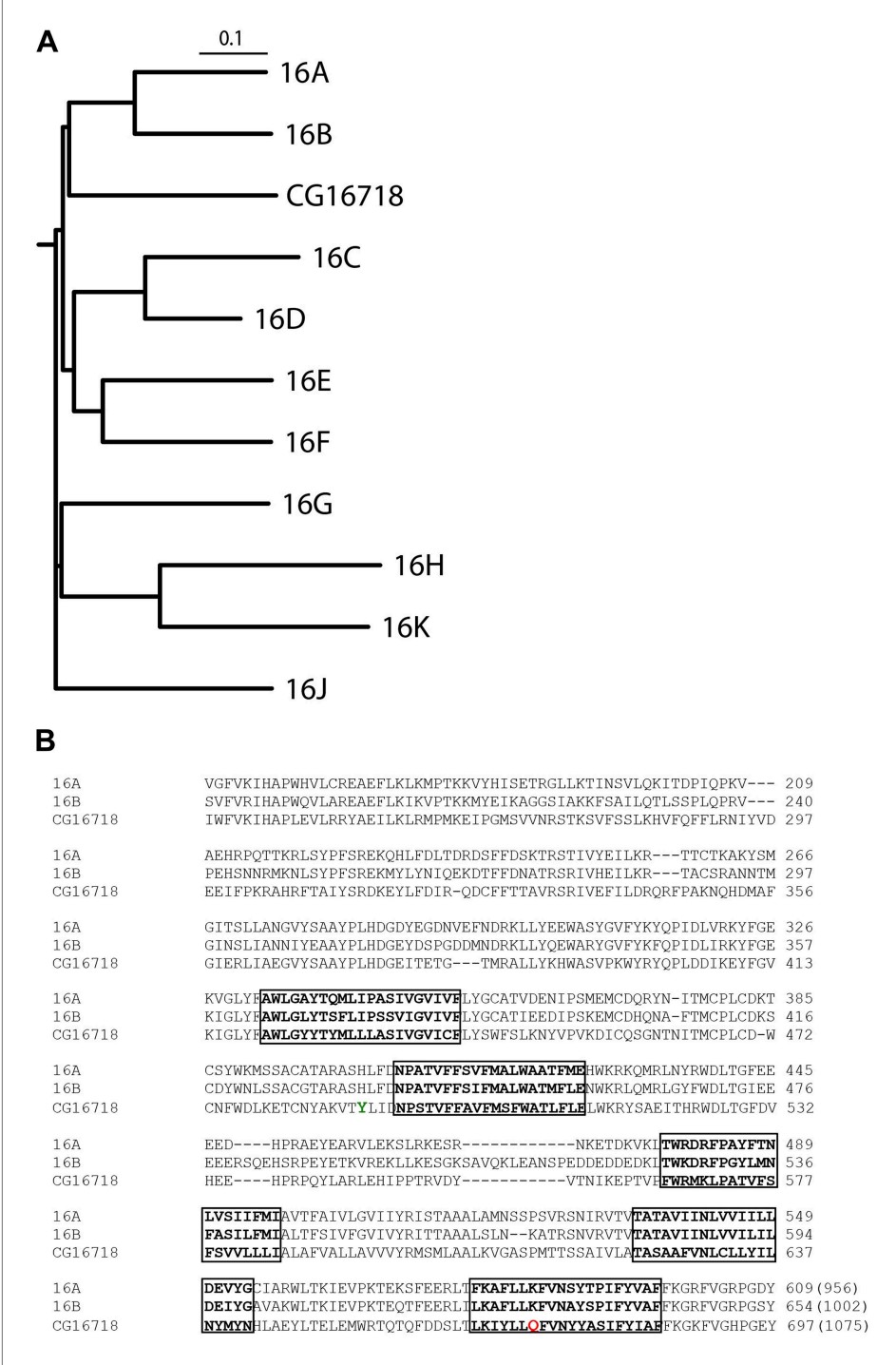

**Figure 1**. CG16718 (Subdued) aligns closely with mammalian CaCCs TMEM16A and B (*Mus musculus*).
(**A**) Multiple sequence alignment of protein sequences was done with ClustalW2 and phylogenetic tree construction
was done in PHYLIP 3.67 (Drawgram). (**B**) Putative transmembrane segments are highlighted with boxes, and
mutated residues (***Figure 4***) featured in this report are marked in color on the primary sequence alignment Y489 is
shown in green and Q672 is shown in red. Bracketed values at the end of the alignment indicate the number
of residues in the whole channel.

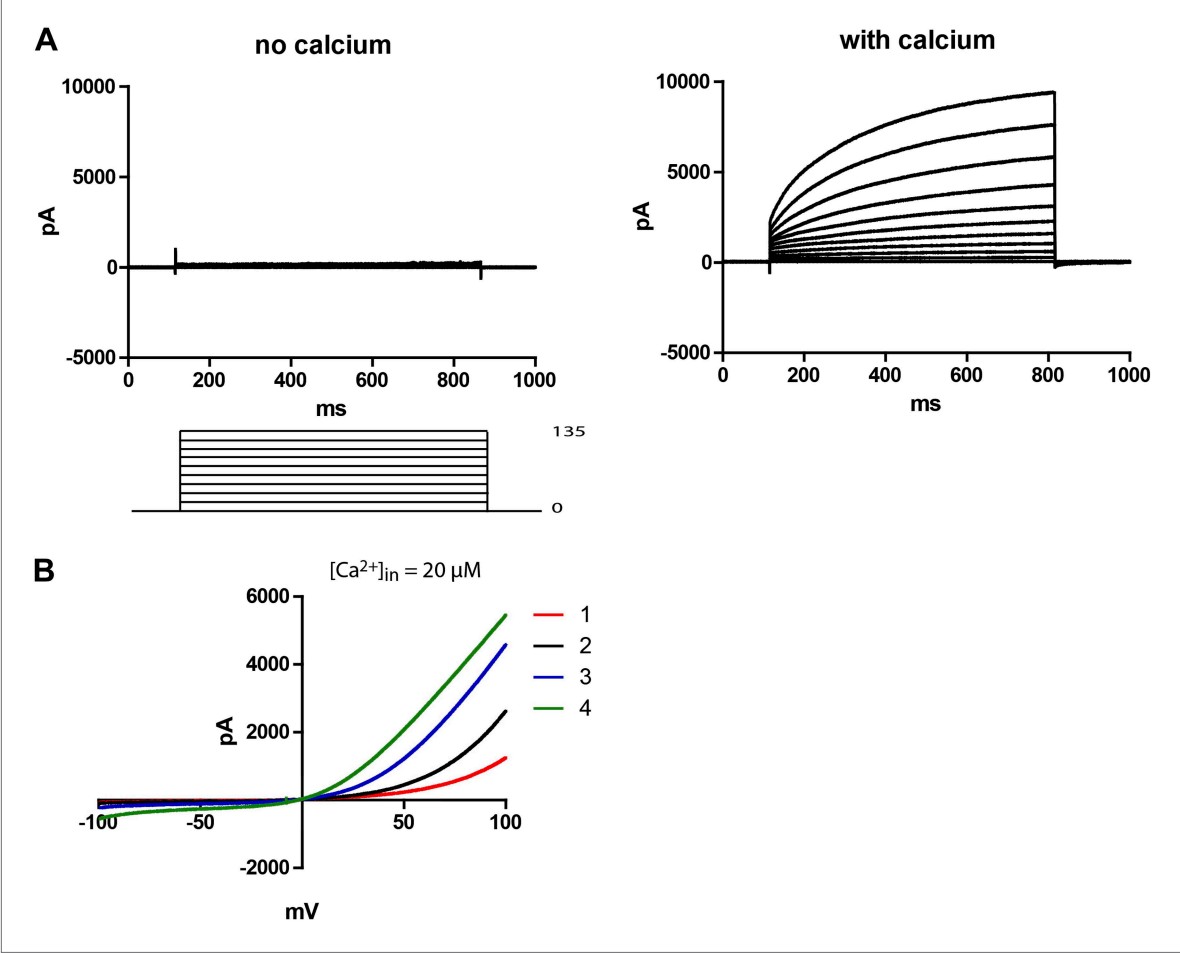

**Figure 2**. Subdued is a calcium and voltage-dependent channel. (**A**) Subdued-transfected HEK 293T cells were used for recording in whole cell patch clamp experiments. No current was observed without calcium in the pipette (left), but large time dependent currents were observed (right) when calcium was added to the pipette (200 μM free calcium). Unless otherwise mentioned, all recordings in this study were done with 200 μM free calcium in the pipette and symmetrical NaCl in the pipette and bath solution. (**B**) Representative traces showing outward rectification of Subdued current in voltage ramps from −100 to +100 mV. Traces 1–4 were taken sequentially and show an increase in current and decrease in rectification over time (20 μM free calcium in pipette). The ramps were done at a rate (dV/dT) of 0.067 V/s.

calcium unless otherwise mentioned. Mock-transfected cells showed little to no CaCC (data not shown). Voltage ramps showed that Subdued was outwardly rectifying, giving rise to more current at depolarized potentials (*Figure 2B*). In these experiments with calcium infused from the whole-cell patch clamp pipette into the cytosol, rectification decreased over time as current amplitude increased. Run-up of current likely results from the process of calcium diffusing from the pipette solution into the cell, suggesting that rectification is also calcium-dependent, as has been shown for CaCCs (*Yang et al., 2008*).

## Subdued primarily permeates chloride

Given that the mammalian TMEM16 family contains both anion and cation channels (*Yang et al., 2012*), we wanted to determine the ionic selectivity of the channel. This was done by varying concentrations of NaCl externally while keeping the intracellular NaCl concentration constant at 140 mM. A series of positive reversal potentials ($E_{rev}$) was obtained upon decreasing the external NaCl concentration, indicating anionic selectivity (*Figure 3A,B*). Using the Goldman-Hodgkin-Katz equation (*Figure 3A* inset) , which describes experimentally obtained reversal potentials as a function of ion concentrations and their respective permeabilities ($P_x$, where X is any ion in the system), $P_{Na}/P_{Cl}$ was calculated to be 0.16, indicating a small permeability for cations, as has been found for mammalian TMEM16A and B (*Pifferi et al., 2009*;

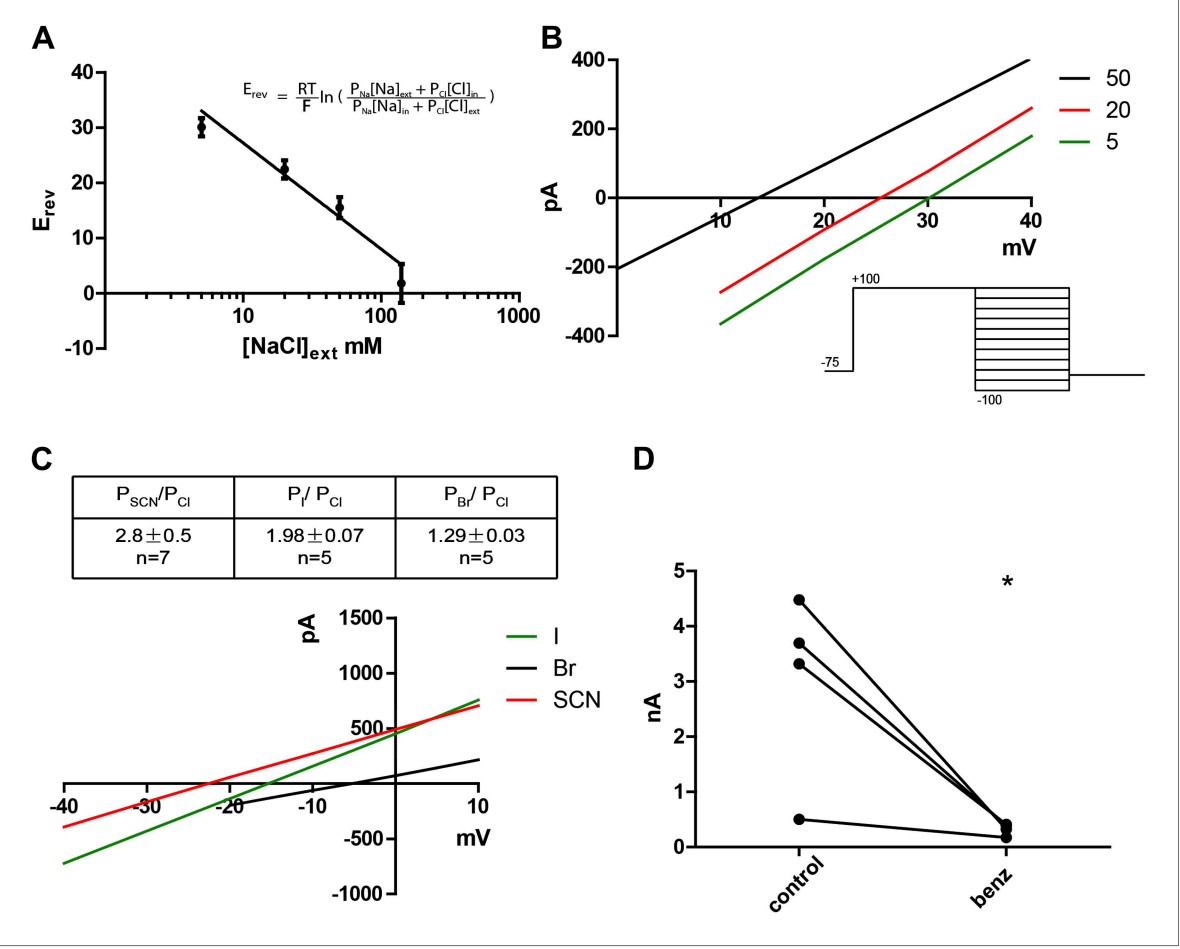

**Figure 3**. Subdued displays hallmark ionic selectivity of classic CaCCs and is blocked by a known CaCC inhibitor. (**A**) NaCl gradients were introduced across the membrane by varying NaCl concentrations of external solutions. The reversal potential ($E_{rev}$) at each concentration was obtained and fitted to the Goldman-Hodgkin-Katz (GHK) equation from which the $P_{Na}/P_{Cl}$ was determined to be 0.16. (**B**) Representative I/V plots obtained by varying external NaCl (in mM). A diagram of the voltage protocol used to measure $E_{rev}$ is shown below. After a 750 ms activating pre-pulse to +100 mV, instantaneous tail currents were measured from test voltages −100 to +100 mV in 20 mV steps. After an initial estimate of $E_{rev}$ using this protocol, test potentials and voltage increments were refined, while pre-pulse conditions and the length of voltage steps remained constant. For each NaCl concentration, n = 4 to n = 9. (**C**) Subdued preferentially permeates larger anions with the selectivity sequence: SCN > I > Br > Cl. Bi-ionic conditions were introduced by varying the external solution. $E_{rev}$ and permeability ratios were obtained by the same methodology as described for (**B**). Representative I/V plots obtained by varying the anion in external solutions using the same voltage protocols as described in (**B**) are shown. (**D**) Subdued is significantly and reversibly blocked by 20 µM benzbromarone (n = 4, p<0.05, Student's *t*-test).

*Yang et al., 2012*). Substitution of chloride with larger halide anions in the external solution revealed that Subdued preferentially permeates the larger anions SCN⁻, I⁻ and Br⁻ relative to Cl⁻ (*Figure 3C*). This hallmark feature of CaCCs implicates hydration energy as a factor in anionic selectivity, a feature shared with cystic fibrosis transmembrane conductance regulator (CFTR) channels (*Qu and Hartzell, 2000*). However, classic CaCC blockers niflumic acid (NFA), flufenamic acid (FFA) and 5-nitro-2-(3-phenylpropylamino)benzoic acid (NPPB) as well as a more recently developed TMEM16A inhibitor T16Ainh-A01 (*Namkung et al., 2011*) did not block the channel (data not shown). Benzbromarone, a TMEM16A blocker identified from a high throughput screen (*Huang et al., 2012a*), blocked Subdued current significantly and reversibly (*Figure 3D*).

## Subdued is a pore-forming subunit

To obtain evidence that Subdued is directly responsible for the observed currents, we introduced mutations onto the channel and observed that the mutant channels produced currents that had different properties from the wild-type channel.

A Q672K mutation produced currents that had significantly slower activation kinetics as compared to wild type (*Figure 4A,B*). Interestingly, the corresponding mutation on mammalian TMEM16F produced a highly similar effect on channel kinetics (*Yang et al., 2012*). A Y489H mutation decreased selectivity for chloride as evident from the decreased shift in $E_{rev}$ upon introduction of a chloride gradient across the membrane (*Figure 4C*). The position of both mutations relative to the first five putative transmembrane domains can be seen on *Figure 1B*, where Y489 is marked in green and Q672 in red. Transmembrane domains are boxed and bolded, and were predicted using the TOPCONS program (*Bernsel et al., 2009*).

### Subdued plays a role in *Drosophila* host defense

FlyAtlas data show that Subdued is expressed at moderate levels in a broad variety of tissues both in larvae and adults, making it difficult to predict a physiological function for this gene in *Drosophila* (*Chintapalli et al., 2007*). From FlyBase-curated data, we noticed that a genome-wide screen for genetic determinants of gut immunity in *Drosophila* reported *subdued* as a susceptibility hit (*Cronin et al., 2009*). In this study, ubiquitous RNAi of *subdued* caused increased lethality upon ingestion of Db11, a particular strain of *Serratia marcescens* isolated from moribund flies (*Flyg et al., 1980*). *Serratia marcescens* is a strain of Gram-negative bacteria that is a common cause of nosocomial infections, and the

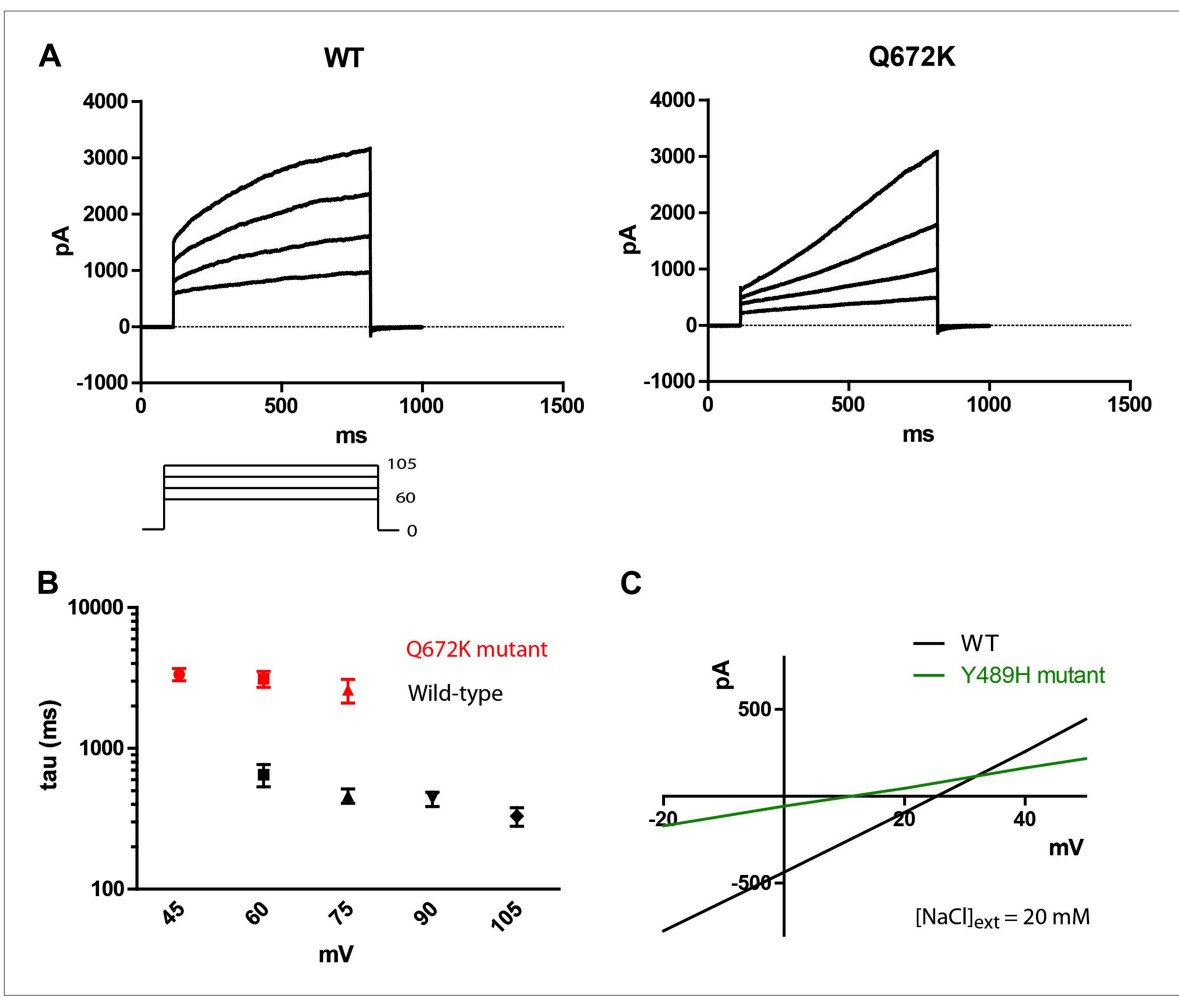

**Figure 4**. Mutations of subdued change properties of observed currents. (**A**) Different kinetic properties in the wild-type (WT) and Q672K mutant channel revealed by a voltage step protocol (750 ms in 15 mV increments). (**B**) Semi-log plots of mean activation time constants (τ) as a function of voltage. τ was derived from the single exponential fitting of the current traces obtained from 750 ms (WT, n = 6) and 5 s (Q672K, n = 4) voltage steps. Time constants at 60 and 75 mV were significantly different for the two channels (p<0.001, Student's *t*-test). (**C**) A Y489H mutation decreases chloride selectivity compared to the WT channel. A representative I/V plot showing the shift in $E_{rev}$ in a 20 mM external NaCl solution. WT $E_{rev}$ was determined to be (25 ± 4) mV, n=5; the Y489H mutant $E_{rev}$ was significantly different at (11 ± 2) mV, (n = 4, p<0.05, Student's *t*-test). Data were obtained using methodology described in *Figure 2B*.

Db11 strain has been shown to be virulent in flies. Importantly, although Db11 kills *Drosophila* in less than 24 hr when introduced via septic injury, its virulence is attenuated when introduced via ingestion (*Nehme et al., 2007*). Survival rates of different fly strains can thus be monitored over a span of a few days, allowing inspection of their relative host defense responses to Db11.

### *subdued* knockout flies are more susceptible to gut infection with *Serratia*

Wild-type flies and two knockout strains generated from independent crosses, KO2 and KO11, were fed Db11 mixed with sucrose solution, and their survival monitored for 8 days, as in previous studies of fly immunity (*Cronin et al., 2009*). Confirming the susceptibility phenotype previously reported, the knockout strains had significantly higher lethality upon being fed Db11 (*Figure 5A*). Over the same timescale as the infection experiment, UV-killed Db11 did not bring about early lethality in any of the strains (*Figure 5—figure supplement 1*). We also determined that *subdued* mRNA was indeed expressed in the gut of wild-type flies but not the genetic knockout strains (*Figure 5—figure supplement 3*).

We hypothesized that since the knockout flies did not display prominent structural abnormalities in the alimentary canal, it was possible that the susceptibility to Db11 infection arose from defects in host defense. At 48 hr post-infection, whole flies were homogenized, and the homogenates were serially diluted and plated on LB agar plates with antibiotic selection. Colony forming units (CFU) were counted on each plate to estimate the number of Db11 bacteria present in the whole fly. Since only live flies were homogenized, this assay reports on the active host response the flies mount against Db11 infection. Significantly higher amounts of Db11 were isolated from knockout flies compared to wild type (*Figure 5B*). To control for the possibility that the higher titers of bacteria isolated from knockout flies could arise from increased feeding, we performed a feeding assay in which a food dye was introduced into the bacteria/sucrose solution (*Ha et al., 2009*). After 72 hr of feeding, flies were dissected to isolate the intact guts and crop, which were homogenized and analyzed for food dye content as a read-out for food consumption. Knockout fly guts did not contain more food dye and surprisingly, slightly but significantly less food dye was recovered from the guts of knockout compared to the wild-type flies (*Figure 5C*). Thus, we surmised that the knockout flies are unlikely to consume more food than wild-type flies. Differences in whole animal bacterial titers are thus likely to result from disparities in host defense. To see if bacteria also accumulated more in the guts as well as the whole animal, homogenates obtained from the feeding assay described above were serially diluted and plated on LB agar plates with antibiotic selection. Knockout fly guts had a significantly higher number of CFU of Db11 than wild-type flies (*Figure 5D*). Since it was reported that injection of latex beads into the hemocoel of Db11-fed flies to impair hemocytic phagocytosis causes significant Db11 proliferation in the hemocoel and adherence of Db11 to the gut wall (*Nehme et al., 2007*), defective hemocoel defenses could also give rise to an apparent increase in the CFU counts from gut dissections. However, homogenate Db11 counts were greatly reduced upon feeding the flies with gentamicin (*Figure 5—figure supplement 2*). This control reveals no significant contributions of hemocoel-resident Db11 to CFU counts from dissected guts. This result also rules out severe hemocoel defense impairment as an explanation for increased in vivo proliferation of Db11.

## Discussion

In this study, we report that Subdued (CG16718), an ortholog of the TMEM16 family in *Drosophila melanogaster*, is a calcium-activated chloride channel with biophysical properties resembling those of classic CaCCs.

This channel is activated by internal calcium, with a lower bound of $[Ca^{2+}]_{in} = 20$ μM in whole cell patch clamp experiments in which current was observed. No significant currents were observed when an EGTA-buffered zero calcium solution was used as the internal solution. Relative to mammalian TMEM16A (*Yang et al., 2008*), TMEM16B (*Pifferi et al., 2009*) or *Xenopus* TMEM16A (H Yang, personal communication, March 2013), Subdued is one to two orders of magnitude less calcium sensitive in whole cell patch clamp experiments. This could either reflect a true biophysical property of the channel or could be an indication that the non-native HEK 293T expression system used in our experiments lacks auxiliary subunits required for higher calcium sensitivity. It would be interesting to test if directed mutagenesis of Subdued can tune its calcium sensitivity within the realm of its mammalian and *Xenopus* counterparts.

Subdued rectifies outwardly, passing larger currents at more positive voltages. The channel permeates mainly chloride with a $P_{Na}/P_{Cl}$ of 0.16, and preferentially permeates larger anions relative to smaller ones, giving the selectivity series $SCN^- > I^- > Br^- > Cl^-$. A Y489H mutation affected the ionic selectivity of the channel, making it more permeable to $Na^+$. This suggests that perhaps the Y489 residue is pore

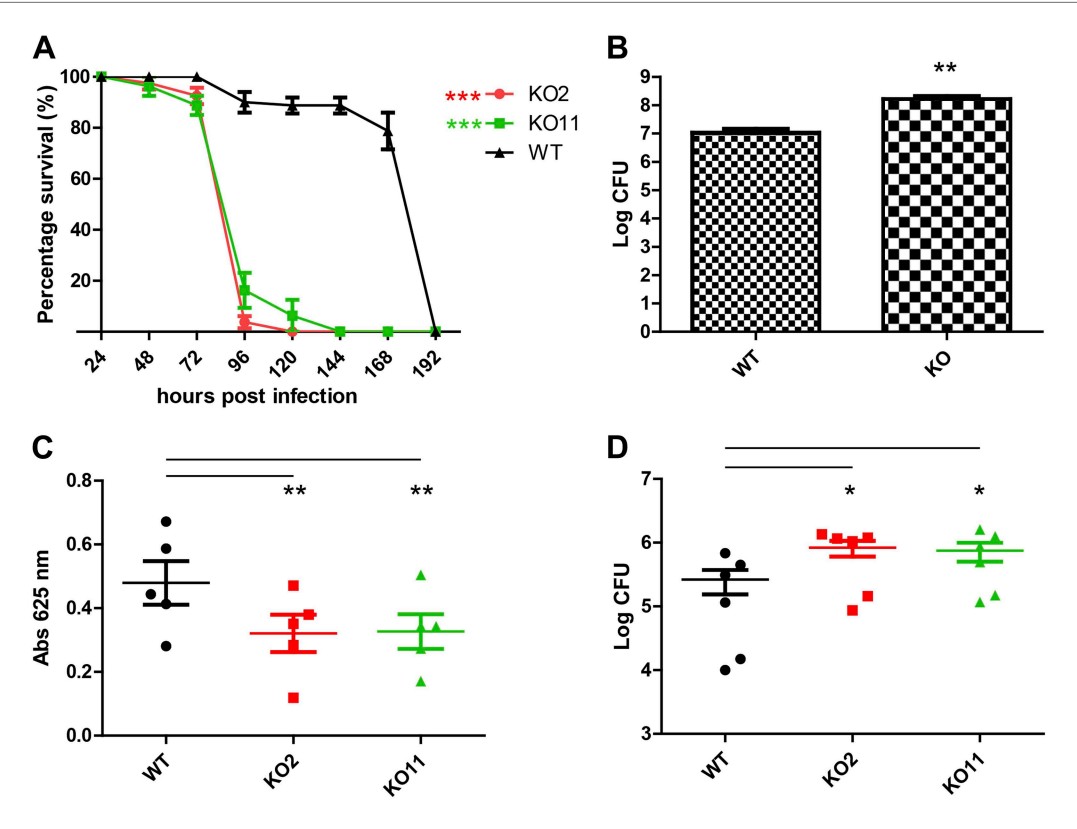

**Figure 5**. Subdued plays a role in host defense in *Drosophila melanogaster*. (**A**) Subdued knockout flies display susceptibility to *Serratia* infection. Wild-type (WT) and knockout (KO) flies were fed on a Db11/sucrose solution and their survival monitored for 192 hr post-infection. Two independently generated KO strains, KO2 and KO11, were used. WT flies lived significantly longer compared to KO2 and KO11 (n = 4, p<0.001, two-way ANOVA). (**B**) KO flies accumulate higher titers of bacteria in the whole animal. 20 whole flies were homogenized 48 hr post-infection. Serially diluted homogenates were plated on agar and inspected for Db11 colony forming units (CFU). Significantly more bacteria were recovered from the KO flies (n = 7, Student's *t*-test, p<0.01). (**C**) KO flies do not consume more food than WT flies. Four fly guts were dissected and homogenized from vials of 20 flies fed with Db11/sucrose solution containing 0.5% wt/vol erioglaucine disodium salt (FDC Blue #1) 72 hr post-infection. The amount of food eaten by the flies was estimated by measuring absorbance of the dye. KO flies tended to consume significantly less food than WT flies (n = 5, p<0.01, repeated measures one-way ANOVA and Tukey's multiple comparison test). No significant difference was observed between the KO strains. (**D**) The homogenate obtained from experiments described in (**C**) was inspected for Db11 colony forming units (CFU). Significantly higher amounts of bacteria were recovered from KO compared to WT fly guts (n = 6, p<0.05, repeated measures one-way ANOVA and Tukey's multiple comparison test). No significant difference was observed between KO strains.

The following figure supplements are available for figure 5:

**Figure supplement 1**. Knockout (KO) flies do not display significant lethality relative to wild-type (WT) flies upon ingestion of UV-killed Db11.

**Figure supplement 2**. Feeding flies gentamicin greatly reduces Db11 counts from gut dissections.

**Figure supplement 3**. *Drosophila* guts express the *subdued* gene.

lining, or has an allosteric effect on the structure of the pore. The Y489H mutant also gave rise to smaller currents compared to the wild-type channel, a reflection of either decreased unitary channel conductance or decreased membrane expression. Additionally, a Q672K mutation produced a dramatic slowing of activation kinetics, a phenomenon also observed when mutating the corresponding residue in mammalian

TMEM16F (*Yang et al., 2012*). The observation that mutations to Subdued alter the properties of the currents strongly points to this protein as a pore-forming subunit of the recorded CaCCs.

Pharmacologically, Subdued is not blocked by the CaCC blockers NFA, FFA, NPPB or T16Ainh-A01. This might arise from structural differences in Subdued, perhaps in the pore, relative to its mammalian and *Xenopus* counterparts. However benzbromarone blocks Subdued significantly and could potentially be used to interrogate the location and properties of the channel pore. In conclusion, as a distantly related TMEM16 family member, Subdued will be useful as a tool in structure/function studies to parse out conserved or divergent biophysical properties such as calcium- and voltage-dependent gating, permeation and ion selectivity.

To study the function of the channel in *Drosophila*, we generated *subdued* knockout strains. Confirming results from a previous genome-wide RNAi study, the knockout was found to be more susceptible to gut infection by a strain of *Serratia marcescens*, Db11 (*Cronin et al., 2009*). An earlier study proposed that the cause of lethality is bacterial proliferation leading to invasion of the gut tissue and subsequent gut distension and escape of bacteria into the hemolymph (*Nehme et al., 2007*).

In the case of the *subdued* knockout, susceptibility arises, at least in part, from deficient host defense, since in vivo proliferation of Db11 was higher in knockout fly guts as well as in the whole animal as compared to wild-type flies. Additionally, we observed that slightly but significantly less food dye was recovered from the guts of the knockout flies. This might arise from lower food consumption by knockout flies, but could also be an indication of increased gut tissue damage due to greater numbers of Db11 in the gut, leading to leakage and diffusion of food dye into the hemolymph. It remains to be determined if higher Db11 titers in the whole animal also result from defective immune responses within the hemolymph. Tissue-specific RNAi of *subdued* using gut or hemocytes drivers did not recapitulate the whole animal RNAi phenotype (*Cronin et al., 2009*), suggesting that Subdued is likely to exert its protective function in a multitude of tissues.

One potential function for Subdued is in the regulation of the secretion of cationic antimicrobial peptides (AMPs), a process that occurs widely on epithelial surfaces and is known to play critical roles in host defense (*Lemaitre and Hoffmann, 2007*). This hypothesis is consistent with the abundant mRNA expression of *subdued* in various epithelial tissues (*Chintapalli et al., 2007*). The susceptibility observed in the *subdued* knockout flies could also be a consequence of a deficiency in dual oxidase (DUOX)–mediated immunity. The DUOX system is reported to be critical in generating reactive oxygen species (ROS) in *Drosophila* gut epithelia (*Ha et al., 2005*). This study reported that strong antimicrobial ROS species are generated by the peroxidase homology domain (PHD) of *Drosophila* DUOX in a chloride-dependent manner. These ROS species are likely to be the highly reactive hypohalites OCl or OSCN, the in vivo production of which requires trans-epithelial anion transport. Additionally, the *Drosophila* DUOX system has also been shown to mobilize downstream of the Gαq-coupled signaling pathway (*Ha et al., 2009*), implicating other calcium-dependent responses in the *Drosophila* immune response. Following Db11 infection of *subdued* knockouts, it is possible that Gαq receptor stimulation fails to elicit sufficient amounts of halide transport onto gut epithelia due to a deficiency in CaCCs, reducing PHD-mediated generation of antimicrobial hypohalites and leading to increased bacterial proliferation and higher lethality.

There remains the possibility that subtle structural deficits also contribute to the susceptibility of *subdued* knockouts to Db11 infection. Developmental defects in gut epithelial integrity (*Bonnay et al., 2013*) or the peritrophic matrix lining the gut (*Kuraishi et al., 2011*) might result in the susceptibility phenotype. The *subdued* knockout flies did not have significant defects in gut epithelial polarity and integrity under basal conditions as assessed by immunostaining for Armadillo and Discs Large (Dlg) to observe adherens and septate junction structure (*Tepass et al., 2001*; *Hortsch and Margolis, 2003*) (data not shown). However, Subdued might function in gut epithelial or peritrophic matrix integrity only upon Db11 challenge to the gut, a possibility that will be explored in future study.

## Materials and methods

### Electrophysiology

CG16718 was subcloned from BDGP *Drosophila* Gene Collection cDNA clone LD10322, which yields the ORF of the RA splice variant of CG16718. Fresh HEK 293T cells (ATCC, Manassas, VA, USA) were transfected with CG16718-eGFPN1 (CG16718 tagged at the C-terminus with EGFP) for 24 hr (Fugene, Madison, WI, USA) and recovered in fresh media for another 24 hr. Patch pipets (World Precision

Instruments, Sarasota, FL, USA) were pulled from a Sutter P-97 puller and re-polished. Pipettes had resistances of 3–5 MΩ for whole cell patch clamp experiments. The bath was grounded via a 3 M KCl agar bridge connected to an Ag-AgCl reference electrode. Data were acquired using a Multiclamp 700B amplifier controlled by Clampex 10.2 via Digidata 1440A (Axon Instruments, Sunnyvale, CA, USA). The standard internal solution contained (in mM) 130 NaCl, 10 HEPES, 5.6 $CaCl_2$, 5 EGTA, 5 MgATP, 1 $Na_2GTP$, 10 phosphocreatine, pH 7.2. The standard external solution was 140 NaCl, 10 EGTA, 2 $MgCl_2$ and 10 HEPES, pH 7.2. The free calcium concentration was calculated to be 200 µM with WEBMAXC software (http://www.stanford.edu/~cpatton/webmaxcS.htm) and was confirmed with a calcium electrode (Orion 4-Star, Thermo Scientific, Waltham, MA, USA). For *Figure 2B*, the internal solution contained (in mM) 130 NaCl, 10 HEDTA, 10 HEPES, 7.55 $CaCl_2$, pH 7.2, with a free calcium concentration of 20 µM. External solutions contained various concentrations (in mM) of 140 NaCl (the default symmetric NaCl condition) or 140 NaX, 10 EGTA, 2 $MgCl_2$ and 10 HEPES, pH 7.2. Sucrose was added to balance osmolarity for the low NaCl solutions. NFA, FFA, NPPB, T16Ainh-A01 and benzbromarone were obtained from Sigma (St. Louis, MO, USA). All experiments were performed at room temperature (22–24°C), and data were analyzed in pCLAMP 10.0 and GraphPad Prism 5.

### *Drosophila* genetics
The CG16718 deficiency line was generated by following closely the heat-shock driven FLP-recombinase methodology previously reported (*Parks et al., 2004*). The fly strains used were PBac{RB}CG16718[e02779] and P{XP}CG16718[d03361] (Harvard Medical School).

### Bacterial infection assays
Flies were reared on standard cornmeal-agar with dead yeast. Infection assays were performed as described in *Cronin et al. (2009)*, with some modifications. Batches of 20 adult flies were used for each line assayed. The food solution containing Db11 was prepared from culture grown exponentially at 37°C in LB (Luria Bertani) medium supplemented with 60 µg/ml ampicillin. This culture was diluted with a freshly prepared sterile-filtered 0.05 M sucrose solution to a final OD (600 nm) = 0.1. Absorbent filters (37 mm; Millipore, Billerica, MA, USA) were thoroughly soaked with the bacteria/sucrose solution and one filter was placed into each vial. 20 flies that were 2 days old were then transferred to each vial, which was then placed at 29°C to start the infection assay. Flies were transferred to new vials with freshly prepared bacteria/sucrose solution every 4 days. $w^{1118}$ flies were used as a control. Data were analyzed with GraphPad Prism 5.

### Colony-forming unit assay
Vials of 20 flies (2–4 days old) were fed with bacteria/sucrose solution as described above. After 48 hr the flies were cold-anesthetized, transferred to eppendorf tubes and homogenized in 500 µl of LB media. After a 4 s spin at 200×g, homogenates were serially diluted into LB, and the serial dilutions plated onto LB-ampicillin plates (60 µg/ml). The plates were incubated at 37°C for 18–24 hr, and the plates with a colony count between 100–500 were chosen for inspection. For gut-specific CFU assays, homogenates from the feeding assay were plated instead. Data were analyzed with GraphPad Prism 5.

### Feeding assay
For the feeding assay, 0.5% wt/vol of erioglaucine disodium salt (Sigma, St Louis, MO, USA) was added to the sucrose solution to be used in the infection assay and the final solution used to dilute Db11 bacteria as described above. After 72 hr of exposure to the bacteria, the flies were cold anesthetized and four female flies were dissected, and the whole gut including the crop was isolated from each fly. The tissue was homogenized in 50 µl of PBS and spun at 8000×g for 10 min. Absorbance of the supernatant at 625 nm was measured using the UV-Vis module of Nanodrop 1000 (Thermo Scientific, Waltham, MA, USA). Data were analyzed with GraphPad Prism 5.

### Reverse transcription (RT-PCR) and quantitative PCR (qPCR)
The SuperScript III First Strand Synthesis System (Invitrogen, Carlsbad, CA, USA) was used for RT-PCR. Random hexamers were used to prime cDNA synthesis.

The following primers were used for qPCR (*Figure 5—figure supplement 3*):

*subdued*—forward: 5′-GCGAATCAATGACTTTGAAC-3′, reverse: 5′-CCCTGGTGATGATTTTGGTG-3′

*β-tubulin*—forward: 5′-ATGAGGGAAATCGTTCACATCCAAG-3′, reverse: 5′-CCCGCGCTGTCCTTGTCGAT-3′

The program used for qPCR is as follows: incubation at 50°C for 2 min and 95°C for 10 min; followed by 40 cycles of denaturation at 95°C for 15 s, annealing and extension at 60°C for 1 min. Dissociation curves of each sample were measured to confirm the specificity of the PCR products.

The following primers were used for regular PCR amplification of cDNA:

*subdued*—forward: 5′-GATCGATTGACCACGGACATTCCTGG-3′, reverse: 5′-CAACGCCCCCACGCTCCAACTGCC-3′

## Acknowledgements

Db11 was a gift from Dr Brian P Lazzaro. We thank Prof Dominique Ferrandon for insightful critique of the manuscript and Tong Cheng for technical support. YNJ and LYJ are HHMI investigators.

## Additional information

### Funding

| Funder | Grant reference number | Author |
|---|---|---|
| Howard Hughes Medical Institute | | Xiu Ming Wong, Susan Younger, Christian J Peters, Yuh Nung Jan, Lily Y Jan |
| National Institutes of Health | NS069229 | Xiu Ming Wong, Susan Younger, Christian J Peters, Yuh Nung Jan, Lily Y Jan |
| Agency for Science, Technology and Research | | Xiu Ming Wong |

The funders had no role in study design, data collection and interpretation, or the decision to submit the work for publication.

### Author contributions

XMW, Conception and design, Acquisition of data, Analysis and interpretation of data, Drafting or revising the article; SY, Conception and design, Acquisition of data; CJP, Acquisition of data, Analysis and interpretation of data; YNJ, Conception and design, Analysis and interpretation of data; LYJ, Conception and design, Analysis and interpretation of data, Drafting or revising the article

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
