## [Decision Letter]

Thank you for sending your work entitled “Subdued, a Calcium-Activated Chloride Channel from the TMEM16 Family in *Drosophila melanogaster*” for consideration at *eLife*. Your article has been favorably evaluated by a Senior editor and 3 reviewers, one of whom is a member of our Board of Reviewing Editors.

The Reviewing editor and the other reviewers discussed their comments before we reached this decision, and the Reviewing editor has assembled the following comments to help you prepare a revised submission.

The reviewers find this work to be well performed and well written. In particular, the characterization of the gene product as a Ca-activated Chloride channel is very convincing. There are substantial issues raised with the immunity angle that need to be addressed through further experimentation for the paper to be acceptable for *eLife*. The following is a summary of the reviewers’ concerns that need to be addressed in a revised manuscript.

1) It is not fully clear whether this channel is involved in resistance or tolerance to infection. While the apparent increase in the gut microbial titer may suggest a resistance function in the gut, quite possibly through DUOX as discussed by the authors, a similar phenotype might be achieved by a defective cellular immune response. It could be that exponentially proliferating bacteria in the hemocoel stick to the outside of the gut and thus lead to the mistaken impression of an increased titer in the gut lumen (see Table S2, [26]). One easy control to test for this possibility would be to feed flies for 24 hr first with Serratia and then with gentamicin and thereafter determine the bacterial titer. If it drops to nil, then there is no systemic infection.

2) A recent publication in Cell from the group of Won-Jae Lee describes a HOCl-specific dye, R19S, to monitor DUOX induction. A simple staining experiment will allow the authors to test their hypothesis about a requirement for Subdued in DUOX function.

3) The authors need to show an important control for Figure 5 feeding flies with killed *S. marcescens* and monitoring survival.

4) It is unclear from the report where the site of action may be. Is the gene product expressed in the gut? An antibody is probably not available, but tissue specific FISH techniques are very sensitive with current technology and can easily demonstrate expression.

5) The authors mention in passing that an RNAi does not give a phenotype in a limited number of tissues tested. This needs to be settled better by A: demonstrating that the existing RNAi indeed functions. And B: using a more extensive tissue specific set of drivers to determine the site of action.

---

## [Author Response]

*1) It is not fully clear whether this channel is involved in resistance or tolerance to infection. While the apparent increase in the gut microbial titer may suggest a resistance function in the gut, quite possibly through DUOX as discussed by the authors, a similar phenotype might be achieved by a defective cellular immune response. It could be that exponentially proliferating bacteria in the hemocoel stick to the outside of the gut and thus lead to the mistaken impression of an increased titer in the gut lumen (see Table S2,*
[26]*). One easy control to test for this possibility would be to feed flies for 24 hr first with Serratia and then with gentamicin and thereafter determine the bacterial titer. If it drops to nil, then there is no systemic infection*.

We have performed the experiment as suggested and found much reduced bacterial titer in the guts of flies fed first with Serratia and then with gentamicin. We added the data as Figure 5–figure supplement 2.

*2) A recent publication in Cell from the group of Won-Jae Lee describes a HOCl-specific dye, R19S, to monitor DUOX induction. A simple staining experiment will allow the authors to test their hypothesis about a requirement for Subdued in DUOX function*.

The suggested DUOX induction experiment was performed as described in the above mentioned paper. Dissecting about 20 guts per experiment (n=3), we observed a statistically non-significant trend (p=0.07, unpaired t-test) of reduced DUOX induction in the knockout flies. A feeding assay showed a slight trend towards higher feeding for the knockout flies (5 guts, n=3) (see Figure 6 below). Thus, even though the knockout flies may have consumed a greater amount of uracil, they appear to display less DUOX induction. However in our hands, high background fluorescence was observed from flies fed with only uracil and not the R19S dye. This occurred despite feeding the flies sucrose solution for a few days after eclosion and pre-starving them for 18 hr on water to reduce auto-fluorescence from normal food. Normalizing to this background, we were only able to observe less than half the number of ROS-positive guts as compared to what has been previously reported for the wildtype control (70%). This technical problem has rendered it difficult to ascertain a statistically significant reduction of DUOX induction by uracil in knockout flies. By retaining a discussion of the hypothesis of a requirement for Subdued in DUOX function, we hope to stimulate future studies that may yield more definitive assessments of this possibility.

**Author response image 1. fig6:** 

*3) The authors need to show an important control for*
Figure 5
*feeding flies with killed* S. marcescens *and monitoring survival*.

We have performed the control experiment as suggested and added the data as Figure 5–figure supplement 1.

*4) It is unclear from the report where the site of action may be. Is the gene product expressed in the gut? An antibody is probably not available, but tissue specific FISH techniques are very sensitive with current technology and can easily demonstrate expression*.

We attempted to raise an antibody, but as the reviewers have anticipated, we could not establish the specificity of the antibody generated. One impediment we encountered while attempting antibody/fluorescence-based methods of establishing the expression pattern and level of *subdued* was the high autofluorescence in the gut, as well as the tendency for non-specific adherence of antibodies to the gut wall. These limitations apply equally to FISH-based methods. We decided to perform RT-PCR and quantitative PCR to show that the gene is indeed expressed in the gut and included the data as Figure 5–figure supplement 3.

*5) The authors mention in passing that an RNAi does not give a phenotype in a limited number of tissues tested. This needs to be settled better by A: demonstrating that the existing RNAi indeed functions. And B: using a more extensive tissue specific set of drivers to determine the site of action*.

To validate the results from the genome-wide RNAi screen performed by Cronin et al., very early on in the project we performed RNAi with commercially available lines, prior to the generation of the CG16718 knockout flies. There were a few issues that were raised by the RNAi data.

Firstly, when we used a ubiquitous Gal4 driver (in this case tubulin-Gal4; Bloomington stock #5138) to drive RNAi we noticed a uniform and prominent curly wing phenotype not present in either parent stock. An actin-Gal4 driver also produced this wing phenotype, but in this case it became impossible to distinguish actin-Gal4/RNAi from actin-Gal4/CyO progeny. The curly wing phenotype was not present in flies bearing deletion of the CG16718 gene and the neighboring CG6231 gene that were generated by crossing two overlapping deficiency lines (Bloomington stocks #7663 and #27,379), nor was the curly wing phenotype observed in the FLP-recombinase generated knockout fly reported in the main manuscript. This raised concerns of either off-target effects of the RNAi construct, or background mutations in the RNAi stock (VDRC #v37472), casting doubt on the reliability of further RNAi experiments.

Secondly, our preliminary experiments showed that tissue-specific-Gal4/RNAi did not seem to have an effect. Gal4 drivers for fat body, hemocyte, gut and Malpighian tubules were used. In an attempt to maximize Gal4 activity, the experiment was repeated with different pre-assay incubation temperatures (25 or 29°C) and times (2 or 4 days) for the flies, but the observed trends were qualitatively the same (representative figure shown below: Figure 7). Multiple enhancer trap lines for expression in different segments of the Malpighian tubules were used but none led to any significantly reduced survival, thus for simplicity a representative Gal4 line (that also drives gut expression) is shown in the figure below. The tubulin-Gal4/RNAi results showed consistent susceptibility relative to either tubulin-Gal4 or RNAi only control, indicating that we might be observing a true phenotype (hence we proceeded to generate a genetic knockout). From tissue-specific RNAi experiments, we surmised that either the gene exerts its protective function in a tissue type we did not anticipate, or that it functions in multiple sets of tissues with redundant functions in host defense, thus knocking it down in only one or two sets of tissues was insufficient to recapitulate the whole body phenotype.

**Author response image 2. fig7:**